# FZD10 Carried by Exosomes Sustains Cancer Cell Proliferation

**DOI:** 10.3390/cells8080777

**Published:** 2019-07-25

**Authors:** Maria Principia Scavo, Nicoletta Depalo, Federica Rizzi, Chiara Ingrosso, Elisabetta Fanizza, Annarita Chieti, Caterina Messa, Nunzio Denora, Valentino Laquintana, Marinella Striccoli, Maria Lucia Curri, Gianluigi Giannelli

**Affiliations:** 1Personalized Medicine Laboratory, National Institute of Gastroenterology “S. De Bellis”, Via Turi 27, Castellana Grotte, 70013 Bari, Italy; 2Institute for Chemical and Physical Processes (IPCF)-CNR SS Bari, Via Orabona 4, 70126 Bari, Italy; 3Dipartimento di Chimica, Università degli Studi di Bari Aldo Moro, Via Orabona 4, 70126 Bari, Italy; 4Laboratory of Clinical Biochemistry, National Institute of Gastroenterology “S. De Bellis”, Via Turi 27, Castellana Grotte, 70013 Bari, Italy; 5Dipartimento di Farmacia, Scienze del Farmaco, Università degli Studi di Bari Aldo Moro, Via Orabona 4, 70126 Bari, Italy; 6National Institute of Gastroenterology “S. De Bellis”, Scientific Direction, Via Turi 27, Castellana Grotte, 70013 Bari, Italy

**Keywords:** colorectal cancer cells, gastric cancer cells, cholangiocarcinoma cells, hepatocarcinoma cells, exosomes, FZD10 protein, FZD10-mRNA, FZD10-mRNA silenced cells, cell proliferation

## Abstract

Extracellular vesicles (EVs) are involved in intercellular communication during carcinogenesis, and cancer cells are able to secrete EVs, in particular exosomes containing molecules, that can be transferred to recipient cells to induce pathological processes and significant modifications, as metastasis, increase of proliferation, and carcinogenesis evolution. FZD proteins, a family of receptors comprised in the Wnt signaling pathway, play an important role in carcinogenesis of the gastroenteric tract. Here, a still unknown role of Frizzled 10 (FZD10) protein was identified. In particular, the presence of FZD10 and FZD10-mRNA in exosomes extracted from culture medium of the untreated colorectal, gastric, hepatic, and cholangio cancer cell lines, was detected. A substantial reduction in the FZD10 and FZD10-mRNA level was achieved in FZD10-mRNA silenced cells and in their corresponding exosomes. Concomitantly, a significant decrease in viability of the silenced cells compared to their respective controls was observed. Notably, the incubation of silenced cells with the exosomes extracted from culture medium of the same untreated cells promoted the restoration of the cell viability and, also, of the FZD10 and FZD10-mRNA level, thus indicating that the FZD10 and FZD10-mRNA delivering exosomes may be potential messengers of cancer reactivation and play an active role in long-distance metastatization.

## 1. Introduction

Extracellular vesicles (EVs) are nanostructures derived from non-plasma membrane and exosomes represent a specific fraction of them that are, according to the established classification, characterized by an average diameter in the range between 50 and 200 nm [1,2]. Significant differences between EVs secreted by normal and cancer cells occur in terms of size and of composition, that can vary depending not only on the intrinsic nature of each individual cell but also on the microenvironment. With respect to the healthy cells, cancer cells produce a higher amount of EVs, and their secretion is modulated by the peculiar extrinsic variables of the tumor milieus, such as increased interstitial pressure, oxidative stress, nutrient deprivation and competition, hypoxia, pH gradients, and growth factors release [3,4,5,6]. Recently several studies proved that cancer cells are able to secrete exosomes that contain molecules, including non-coding RNA, miRNAs, mRNA, and proteins that can be transferred to recipient cells to induce new biological processes, thus causing significant modifications as angiogenesis, therapeutic resistance, formation of metastasis, and an increase of proliferation activity [7]. In our previous study we demonstrated the involvement of small EVs derived from human plasma in the carcinogenetic signal transmission in colon and gastric cancer. In particular, the expression level of Frizzled-10 (FZD10), a protein of the Frizzled family, found, for the first time, to be specifically localized in the small EVs, was demonstrated at a higher level in the oncological patients than in the healthy control group [8,9]. The Frizzled family, a distinct group within the superfamily of G protein-coupled receptors, consists of ten different seven-transmembrane receptors groups, divided into four clusters based on their amino-acid composition. FZD10 protein is a cell surface receptor, which is activated by Wnt proteins and involved in the regulation of cellular function [10,11,12,13,14,15] and belongs to the cluster 3, along with FZD4 and FZD9 [16]. We proved that the modulation of the FZD10 expression level in small EVs strongly depends on the stage of disease, and that, consequently, the evolution of pathology can be monitored by evaluating the level of protein expression therein [8].

In different cancer cell lines such as colon, gastric, hepatic, and cholecystic cancer cell lines, a modulation of FZD10 protein expression depending on mRNA modulation [17] was demonstrated; however, very few evidences were found on the role of the FZD10 containing small EVs during the carcinogenesis or even, more in general, of the Frizzled proteins. Nagayama et al. found involvement of FZD10 in synchronous colorectal tumors, indicating the FZD10 contribution to the non-canonical Wnt signaling pathway [18]. 

Here, an in vitro study was performed on exosomes secreted by different cancer cell lines derived from the gastro enteric tract. FZD10-mRNA silencing proved that there was a reduction in the proliferative index only for specific tested cell lines that reported a reactivation of cell viability upon exposure to the exosomes secreted by the corresponding untreated cell line. Furthermore, the investigation of the specific genetic signal in such cell lines, and in the corresponding exosomes, demonstrated that the FZD10-mRNA expression level, also reduced upon silencing, was restored after incubation with the exosomes of the untreated cell lines, reaching values comparable to those present in the pristine cells and their exosomes [19,20,21,22,23,24,25]. The study aimed to investigate the role of FZD10 on different cancer cell lines during the proliferation.

## 2. Materials and Methods

### 2.1. Cell Lines and Cultures

Human colorectal adenocarcinoma cells (CaCo-2), metastatic SW-620 colon cancer cells, hepatocellular carcinoma cell lines (Hep-3B, HLF and HLE cells), liver hepatoma cells (PLC-5), and gastric carcinoma (derived from metastatic site) N-87 cells were purchased from ATCC. Human gastric carcinoma (derived from metastatic site) cell line (HGC-27) was purchased from Sigma-Aldrich. Human intrahepatic cholangiocellular carcinoma cell line (HUCCT-1) was purchased from JCRB Cell Bank derived from ascitic fluid. All cell lines were cultivated in according to retailer protocols. Briefly, colon cancer cell lines (CaCo-2 and SW-620) were cultured in 10% of Inactivated Fetal Bovine Serum (FBS) exosomes free (Euroclone) in Dulbecco’s Modified Eagle’s Medium (DMEM) with sodium pyruvate, 4.5 g/L glucose (GIBCO), 4mM L-Glutamine (GIBCO) and 5 mL Pen-Strep (penicillin 10,000 u/mL, streptomycin 10,000 u/mL, Lonza Biowhittaker). Gastric (N-87 and HGC-27), hepatocellular carcinoma (Hep-3B, HLE and HLF), hepatoma (PLC-5), and intrahepatic cholangiocellular carcinoma (HUCCT-1) cell lines were cultured with Roswell Park Memorial Institute (RPMI) medium (GIBCO) supplemented with 10% of FBS exosomes free, sodium pyruvate, 4.5 g/L glucose, 4mM L-Glutamine, and 5 mL Pen-Strep.Cells were grown until reaching semi-confluence, in a humidified incubator at 37 °C with an atmosphere containing 5% of CO_2_.

### 2.2. FZD10-mRNA Silencing and Cell Viability Assessment

Nine different cell lines (HGC-27, HLF, HUCCT, N-87, SW-620, CaCo-2, Hep-3B, PLC-5, HLE) were seeded into sterile 96-well culture plates at a density of 2 × 10^3^ cells/well. After 24 h, cells were silenced for 96 h by using si-PORT-NeoFX transfection agent (Thermo Fisher Scientific) and silencer select FDZ10-siRNA (Thermo Fisher Scientific), according to the protocol provided by Thermo Fisher Scientific. Untreated adherent cells were trypsinized and diluted in normal growth medium for 1 h before transfection. Cells were set aside at 37 °C in incubator while transfection complexes were prepared in sterile tubes. siPOR-NeoFX transfection agent (0.5 µL) was diluted in Opti-MEM I (10 µL) and incubated for 10 min at room temperature. Similarly, silencer select FDZ10-siRNA was diluted in Opti-MEM medium at room temperature to a final concentration of 5 nM. The formation of the transfection complexes was achieved by mixing the diluted solution of siPORT-NeoFX transfection agent and FDZ10-siRNA. The resulting mixture was incubated for 10 min at room temperature and then used to overlay the cells. The culture plates were gently tilted to achieve a homogeneous mixing. Cells treated only with diluted solution of siPOR-NeoFX transfection agent, keeping unchanged all other experimental conditions, were used as negative controls for each tested cell line. Cell viability was investigated by performing MTS cell proliferation assay (CellTiter 96 AQ_ueous_ One Solution Cell Proliferation Assay MTS, Promega). After 96 h of incubation, 20 μL of MTS tetrazolium compound were added to each well and the plates were incubated for an additional 3 h at 37 °C. The absorbance of the FDZ10-mRNA-silenced cells and of the negative control cells was measured at 490 nm using a Perkin Elmer Victor Plate Reader (Zaventem, Belgium).

### 2.3. Exosomes Extraction

Cells lines were grown in a sterile T25 culture flask (5 mL). Exosomes were extracted starting from culture medium collected only by the cell cultures of those cell lines that were found to be responsive in terms of viability to the FDZ10-mRNA silencing experiment, before and after the treatment. The exosomes were extracted by following the procedure previously reported by R. J. Lobb et al. [26]. In particular, for the exosomes extraction the culture medium of cells was centrifuged for 10 min at 1500× *g* (Thermo Scientific, Heraeus Multifuge X3 Centrifuge). The supernatant was transferred to a clean tube and centrifuged again at 1800× *g* for 10 min, after which it was carefully transferred into a sterile tube (15 mL). The medium was then further centrifuged at 3000× *g* for 15 min and the supernatant was transferred into a clean tube for another centrifugation at 3800× *g* for 15 min. Subsequently, the supernatant was ultra-centrifuged at 75,000× *g* for 2 h (BECKMAN, L-60 Ultracentrifuge), and, after its separation from pellet, again ultra-centrifuged at 100,000× *g* for 2.5 h. All centrifugation steps were carried out at 4 °C. Finally, the pellet formed of exosomes was recovered and dispersed in 200 µL of ultrapure water. The same experimental procedure was used for the extraction of exosomes derived from HGC-27, SW-620, N-87, HUCCT-1, and HLF cells after FDZ10-mRNA silencing experiment. The exosomes were then ready for their characterization, protein extraction, or incubation with cells. The extracted exosomes were stored at -80 °C until proteins analysis was performed. The extraction of total protein content from exosomes was carried out on homogenized samples. For the TEM investigation, 5µL of aqueous suspension of freshly extracted exosomes were cast onto an amorphous carbon-coated Cu grid (CF400-CU-TH, 50/pk, Electron Microscopy Sciences). After sample drying, positive and negative staining was performed before exosomes observation by TEM.

### 2.4. Restoration of Cell Viability by Treatment with Exosomes

HGC-27, SW-620, N-87, HUCCT-1 and HLF cells were seeded into sterile 96-well culture plates at a density of 2 × 10^3^ cells/well. Negative controls and FDZ10-mRNA silenced cells were obtained by following the experimental procedure described in the Appendix A. After 96 h of incubation with the transfection complex, each FDZ10-mRNA silenced cell line was further incubated with the corresponding extracted exosomes, containing a total protein concentration of 20 µg/µL, either with or without the transfection complex. The exosomes extracted from the culture medium of the untreated cells, for each tested line, were used for this experiment. After 96 h, cell viability was evaluated by performing MTS cell proliferation assay, according to the experimental procedure reported above.

### 2.5. RNA Extraction and Real Time Polymerase Chain Reaction

Quantitative real-time polymerase chain reaction (PCR) was performed on c-DNA, derived from FDZ10-mRNA silenced HGC-27, HLF, HUCCT-1, N-87, SW-620 cells, their corresponding negative controls and exosomes, as well as from FDZ10-mRNA silenced cells after their subsequent incubation with exosomes, either with or without the transfection complex. Total RNA was extracted using the miRNeasy Kit in according to the experimental procedure from Qiagen, and after RNA extraction, the purity and the quantity of the nucleic acids were measured with a NanoDrop UV spectrophotometer (Thermo Fisher Scientific, Wilmington, DE, USA). 2 μg of total RNA were reverse transcribed by using High Capacity cDNA Reverse Transcription Kit (Applied Biosystems). Quantitative real time PCR (qPCR) was carried out by means of iTaq™ Universal SYBR^®^ Green Supermix (Bio-Rad) and the CFX96 Touch™ qPCR System (Bio Rad). The optimized thermal cycling conditions were 95 °C for 2 min, 40 cycles at 95 °C for 5 s and 60 °C for 30 s. Primer sequences: GAPDH FW 5’ GAAGGTGAAGGTCGGAGTCA 3’, GAPDH RV 5’ CATGGGTGGAATCATATTGGA 3’; FZD10 FW 5’ AGCAGGTCTCTACCCCCATC 3’, FZD10 RV 5’ TAATCGGGGAGCACTTGAGC 3’. Real-time PCR results were extrapolated from a standard curve and expressed as target sequence copy number per 1 μL c-DNA.

### 2.6. Proteins Extraction and FDZ 10 Quantification by Western Blotting

For HGC-27, SW-620, N-87, HUCCT-1 and HLF cells, the corresponding exosomes, the FDZ10-mRNA silenced cells, before and after the restoration of cell viability experiment, and the negative controls were lysated by using 1× radio immunoprecipitation buffer (RIPA; Cell Signaling Technology, Danvers, MA, USA) containing protease inhibitor (Amresco, Solon, OH, USA), and the total proteins content in the lysate was measured by means of Bradford kit assay (Bio-Rad Hercules, CA, USA). An equal amount of proteins, extracted from the cells of each lines samples (20 µg), was mixed with reducing Laemmli-buffer, loaded on 4–15% Tris-glycine sodium dodecyl sulfate-polyacrylamide gels (Bio-Rad, Hercules, CA, USA) and electrophoresed. Subsequently, proteins were blotted to nitrocellulose membranes (Bio-Rad, Hercules, CA, USA) using Trans-blot system (Bio-Rad, Hercules, CA, USA). The blotted membranes were treated with 5% non-fat milk (Bio-Rad, Hercules, CA, USA) in Tris-buffered saline supplemented with 0.05% Tween-20 (TBS-T) for 1 h, to block non-specific binding sites and then were incubated with primary antibodies, namely anti-FZD10 (1:400; Abcam, Cambridge, UK), anti-GAPDH (1:1000 Abcam, Cambridge, UK). After three washing with TBS-T, membranes were incubated with corresponding HRP-conjugated secondary antibodies (1:1000 Santa Cruz, Santa Cruz, CA, USA) for 1 h at room temperature and subsequently washed with TBS-T. The chemiluminescence signal from proteins was imaged after incubation by using an enhanced chemiluminescence kit (Bio-Rad, Hercules, CA, USA) by Chemidoc XRS+ (Bio-Rad, Hercules, CA, USA). The images were analyzed by using Image Lab 5.2.1 software. The same procedure was used to analyze the presence of FZD10 and of the exosomes markers in the exosomes extracted from cell media supernatant before and after the silencing experiments. Specifically, anti-ALIX (1:500 Santa Cruz, Santa Cruz, CA, USA), anti-HSP-70 (1:500 Santa Cruz, Santa Cruz, CA, USA) and anti-CD63 (1:500 Abcam, Cambridge, UK) were used.

### 2.7. Immunofluorescence Staining of FDZ 10 in Cells

All cells lines were seeded into 6-well plates at a density of 5 × 10^4^ cells/well at 37 °C. The immunofluorescence analysis was performed to detect the FDZ10 expression level in the FDZ10-mRNA silenced cells, in the FDZ10-mRNA silenced cells after treatment with exosomes, in presence and in absence of transfection complex, and in the negative controls, respectively. Subsequently, cells were washed with PBS, fixed with cold ethanol for 30 min at 4 °C and permeabilized for 15 min with 0.5% Triton X-100 in PBS. Then the cells were blocked with 5% normal serum in PBS for 1 h at room temperature and then incubated at 4 °C with the primary antibody against FZD10 (rabbit polyclonal anti-FZD10 from abCam, Cambridge) over-night. The treated cells were then incubated with a specific green-fluorescent conjugated secondary IgG Alexa 488 (Invitrogen) for 1 h and mounted using prolong gold antifade reagent containing 4’,6-diamidin-2-fenilindolo (DAPI, Vector). Images were acquired and analyzed by using the Eclipse Ti2 by Nikon fluorescence microscope and the Interactive software installed on the machine. The images were acquired by using a Kr-Ar and Ar lasers with 488 nm and 358 nm band-pass filters, respectively, for the FZD10 green channel (488 nm) and for the DAPI blue channel (358 nm), at 40× magnification. The fluorescence intensity was quantified by using an exposure time of 90 milliseconds per each acquisition for all the investigated samples.

### 2.8. Immuno-detection of FDZ 10 in Exosomes by Transmission Electron Microscopy (TEM)

3,4-dimethylbenzenethiol capped Au nanoparticles (Au NPs) were synthesized by following a reported procedure [27], dispersed in chloroform (8 × 10^−5^ M) and deposited by drop casting (5 µL) onto a TEM carbon-coated copper grid (CF400-CU-TH, 50/pk, Electron Microscopy Sciences) letting the solvent evaporate. After sample drying, the Au NPs cast onto the grid were incubated with 100 µL of 2 % (v/v) of anti-FZD10 antibody (anti- FZD10, 1:400; Abcam, Cambridge, UK) solution in PBS (10 mM at pH 7.4) for 16 h at 4 °C. After washing with PBS, the resulting Au NPs conjugated with anti-FZD10 onto the grid were further incubated with 100 µL of 0.l% (w/v) of BSA (Sigma-Aldrich) solution in PBS for 1 h at 4 °C, to block non-specific binding [28,29,30]. After washing with PBS, the sample on the grid was incubated with 10 µL of exosomes dispersed in water (concentration of total protein 20 µg/mL) for 15 min and, subsequently, carefully washed with ultrapure water. Ultrapure water obtained from a Milli-Q gradient A-10 system (Millipore, 18.2 MΩ cm, organic carbon content ≥4μg/L) was used to prepare all of the aqueous solutions. After complete drying of the samples, positive or negative staining were performed preliminarily to TEM investigation.

### 2.9. Transmission Electron Microscopy Investigation

The morphological characterization of exosomes and Au NPs was performed by TEM using a Jeol JEM-1011 microscope, working at an accelerating voltage of 100 kV and an Olympus Quemesa Camera (11 Mpx) was employed to acquire the images. The positive staining was accomplished by dipping the grid into a freshly prepared solution of 2% (w/v) phosphotungstic acid in ultrapure water for 3 s. Then, the grid was washed by using ultrapure water to remove the excess of staining agent. The sample was then left to dry overnight and stored in a vacuum chamber until TEM observation. The negative staining was achieved by casting 5 µL of a freshly prepared solution of 2% (w/v) phosphotungstic acid in ultrapure water on the sample deposited on the grid and left for 30 or 60 s. Ultrapure water was used to eliminate the excess of staining agent. After complete evaporation of water, the dried sample was kept in a vacuum chamber until TEM investigation. 

### 2.10. Dynamic Light Scattering (DLS) and ζ-Potential Investigation

Size distribution, stability, hydrodynamic diameter of extracted exosomes, and corresponding polydispersity index (PDI) were evaluated by means of the Zetasizer Nano ZS, Malvern Instruments Ltd., Worcestershire, UK (DTS 5.00). Surface charge of exosomes was investigated by performing ζ-potential measurements recorded by means of a laser Doppler velocimetry (LDV). All data were reported as average values ± standard deviation, considering three replicates.

### 2.11. Statistical Analysis

The Sigma Stat 3.1 software was used for statistical analysis. Statistical significance between two groups was assessed using the Student’s t-test (unpaired), and multiple comparisons were performed by using one-way analysis of variance. When the hypothesis of the mean equality among groups was rejected by the one-way analysis of variance, the Kruscall-Wallis test was applied. We considered statistically significant a difference between the results for each cell line and those of the corresponding untreated cells for either *p* < 0.005 and *p* < 0.001.

## 3. Results

### 3.1. FZD10-mRNA Silencing Experiment in Cell Culture

The effect of FZD10-mRNA silencing on the cell proliferation activity was investigated evaluating cell viability by MTS cell proliferation assay of FDZ10-mRNA silenced CaCo-2, HGC-27, SW-620, N-87, HLE, HLF, Hep3B, PLC-5, and HUCCT-1 lines and the results are reported in Figure 1. Cells treated with siPOR NeoFX transfection agent were only used as negative controls for each tested line. After FDZ10-mRNA silencing, a significant reduction of cell viability was recorded for HGC-27, SW-620, N-87, and HUCCT-1 cell lines, respectively (Figure 1). Conversely, cell viability of CaCo-2, HLE, Hep3B, and PLC-5 cells was not affected at all by the silencing experiment. The obtained data clearly indicated that a statistically significant reduction in cell viability was induced by FDZ-10-mRNA silencing only on HGC-27, SW-620, N-87, HUCCT-1 ((**) *p* < 0.001 versus negative control) and HLF cells ((*) *p* < 0.005 versus negative control).

### 3.2. Exosomes Isolation and Characterization and Restoration of Cell Viability

Exosomes were extracted only from the culture medium collected from the untreated HGC-27, SW-620, N-87, HUCCT-1, and HLF cells, as only their viability resulted in being significantly affected by FZD10-mRNA silencing experiment. TEM, DLS, and ζ-potential investigation of the exosomes freshly isolated from culture medium of the selected cell lines elucidated their morphology, size and size distribution, as well as surface charge. The representative TEM micrograph of the positively stained exosomes extracted from culture medium of HUCCT-1 cells (Figure 2A) highlights the presence of round objects with a size ranging from 30 to 150 nm. The micrographs of the exosomes obtained upon negative staining clearly point out the occurrence of structures compatible with cup-shaped membrane vesicles, delimited by a lipid bilayer (Figure 2B, B1 and B2). In Figure B1 and B2 close-ups of the exosomes of 53, 95, and 138 nm are shown. Similar results were obtained for HGC-27, SW-620, N-87, and HLF cells (Appendix A). The average hydrodynamic diameter, measured by DLS investigation for HGC-27, SW-620, N-87, HUCCT-1, and HLF cells and reported in Figure 2C, resulted in lower than 200 nm and, therefore was in agreement with the result of the TEM observation. ζ-potential measurements (Figure 2C) revealed a negatively charged surface of the exosomes, for each investigated cell line, as expected for cell membranes based on phospholipids with their negatively charged phosphate groups.

HGC-27, SW-620, N-87, HUCCT-1 and HLF cells were then selected for the cell viability restoration experiment performed by using the exosomes extracted by the not-silenced corresponding cell lines. The results of the MTS assay (Figure 3) revealed that, after incubation for 96 h with a mixture of transfection complex and exosomes, all of the investigated FDZ-10-mRNA silenced cell lines restored their viability to a different extent compared with the corresponding negative controls. Conversely, a complete recovery of viability was observed for all tested cell lines for the treatment with the corresponding exosomes only. Interestingly, an increase in cell viability far beyond 100% versus negative control was recorded for N-87 cells only, irrespectively of the applied treatment (Figure 3).

### 3.3. Quantification of FZD10-mRNA Expression Level by Quantitative Real Time PCR

The FZD10-mRNA expression level in the exosomes of the untreated cells and of the FZD10-mRNA silenced HGC-27, SW-620, N-87, HLF, and HUCCT-1 cells (Figure 4A) was investigated. After the silencing experiment, a significant decrease in FZD10-mRNA relative fold change was observed in the exosomes extracted from the different cell lines (Figure 4A).

Similarly, the FZD10-mRNA expression level evaluated by q-PCR in silenced cell lines, before and after the same treatments was found to significantly reduce for all investigated cell lines (Figure 4B).

A complete restoration of FZD10-mRNA expression level was attained after treatment of silenced cells with exosomes only for each investigated cell line (Figure 4B). Furthermore, the silenced HLF, HUCCT, and N-87 cells reached a complete restoration of their FZD10-mRNA expression level after their incubation with exosomes even in presence of the transfection complex (Figure 4B). While the silenced SW-620 and HCG-27 cells exposed to the mixture of transfection complex and exosomes are still a complete restoration of the FZD10-mRNA, expression level was not reached (Figure 4B).

### 3.4. Quantification of FZD10 Expression Level by Means of Western Blotting and Immunofluorescence Imaging

The Western Blotting performed on the exosomes isolated from cultured medium of negative control and FDZ-10-mRNA silenced cells reveals in all the investigated samples the presence of bands ascribable to FZD10. Bands assigned to ALIX, HSP-70 and CD63 proteins, that are established exosomal protein markers, are also evident (Figure 5A). Semi-quantitative immunoblotting carried out by videodensitometry, allowed us to evaluate the average FZD10 expression level in all the exosomal extracted proteins samples, normalized by using the corresponding housekeeping HSP-70 protein bands (Figure 5B). Such semi-quantitative evaluation proved a significant reduction in FZD10 expression level in the exosomes isolated from the silenced cells with respect to that measured in the exosomes isolated from untreated cells, for each investigated cell line (Figure 5B).

Western Blotting analysis carried out on the FDZ 10-mRNA silenced HGC-27, SW-620, N-87, HLF, and HUCCT-1 cells, after their incubation with exosomes, in presence or absence of the transfection complex, and on the corresponding negative controls, confirmed the presence of the FZD10 also in the cells (Figure 6A). A semi-quantitative analysis of the average FZD10 expression level in the cell extracted proteins samples, normalized by using the corresponding housekeeping GAPDH bands for each of the cell lines (Figure 6B) interestingly resulted in a significant reduction of the protein expression level in the silenced cells; while a restoration of the protein expression in the cell treated simultaneously with the transfection complex and the exosomes or with the exosomes only after the silencing experiment (Figure 6B) was observed.

Immunofluorescence imaging was also performed to monitor changes in the FZD10 protein expression in the silenced cells, for the samples obtained after the different treatment in respect to their negative controls. Confocal microscopy images of the negative controls and silenced cells, before and after cells incubation with a mixture of transfection complex and exosome and with exosomes only are reported in Figure 7A–E. The cellular immunofluorescence images demonstrate, for each tested line, a prominent reduction in green fluorescence intensity in the silenced cells, with respect to the corresponding negative controls, thus suggesting a significant decrease of the FZD10 expression level. Furthermore, the reduced blue fluorescence intensity due to the cell nuclei labeling indicates an extensive decrease in cell number for each investigated cell line, thus confirming the results of the cell proliferation tests.

After treating each cell line simultaneously with the mixture of transfection complex and exosomes, or with exosomes only, the confocal microscopy images recorded after at each treatment, along with the corresponding immunofluorescence intensity index histogram, clearly demonstrated comparable or increased green fluorescence intensity with respect to the negative control. Similarly, an increase of the nuclei blue fluorescence is found. The restoration of FZD10 expression level, as well as of the cell viability, was confirmed to take place upon incubation of the silenced cells with the exosomes, either in presence or absence of the transfection complex, in each investigated cell line.

In addition, the immunofluorescence assay allows us to also localize FZD10 on the cell membrane, in agreement with the reports on Frizzled receptor family.

### 3.5. Immunodetection of FZD10 by TEM Analysis

A TEM grid was used as a promptly available support for the deposition of Au NPs functionalized with FZD10 protein primary antibody to probe the presence of the protein at the surface of the exosomes. In particular, suitably synthesized organic capped Au NPs of (2.3 ± 0.5) nm were cast on carbon coated copper TEM grid, that was firstly incubated with an aqueous solution of FZD10 primary antibody to promote the interaction with NPs surface, then with an aqueous solution of BSA and finally with exosomes dispersion in water. Figure 8 reports representative TEM micrographs obtained after positive staining of (A) the exosomes freshly extracted from the culture medium of HLF cells and (B) 3,4-dimethylbenzenethiol capped Au NPs, respectively. The exosomes extracted from the culture medium of HLF cells (Figure 9) appear able to bind the FZD10 antibody functionalized Au NPs. The treatment with BSA solution prevented non-specific binding of the exosomes to the surface of FZD 10 antibody functionalized Au NPs. In fact, the exosomes were found not to bind to *as synthesized* Au NPs cast after treatment with BSA for 1 h and then exposed to the exosomes (Figure 8C), or treated with the exosomes only (Figure 8D). The molecular antigen/antibody recognition occurred only when AuNPs were functionalized with FZD10 antibody that could, thus, effectively bind the FZD10 protein present at the exosomes surface membrane. The same results were obtained for the exosomes extracted from HGC-27, SW620, N-87, and HUCCT-1 (E) cells ((Appendix A)).

## 4. Discussion

During cell carcinogenesis and embriogenesis, cell-cell communication is ensured by interactions through cell junctions, which are multiprotein complexes that provide contact between neighboring cells and between cells and the extracellular matrix [31]. Extracellular vesicles are involved in the different physiological and pathological processes that cells exploit to achieve a long-distance cell-cell communication. Indeed, extracellular vesicles are also defined as “communicasomes” [32]. The first evidence regarding the involvement of “communicasomes” in the carcinogenesis dates back to about 15 years ago [1], and since then several indications of the content of extracellular vesicles in terms of nucleic acids and proteins were attained.

Here, an original and still unidentified role of FZD10 protein was unveiled. So far, this protein was considered a receptor of Wnt membrane complex. In a previous study carried out on tissues, FZD10 protein was demonstrated to be involved in the colorectal and gastric carcinogenesis according to an intracellular migration, from the nucleus to cytoplasm, and from cytoplasm to cytoplasmic membrane, during the cancer development [9], as well as in synchronous colorectal cancer. Very recently, we demonstrated that FZD10 protein is delivered solely by the small EVs extracted from the plasma of patients affected by CRC and GC, both primary and metastatic index of disease [8]. Here, for the first time, the presence of FZD10 and FZD10-mRNA in exosomes, *i.e*. small EVs in a defined size range, extracted from untreated HGC-27, SW-620, N-87, HLF, and HUCCT-1 cells, was detected.

A remarkable decrease in FZD10 and FZD10-mRNA content was observed in the silenced cells and in their corresponding extracted exosomes, thus indicating that the FZD10 message can be communicated by the cells by means of exosomes. Furthermore, a statistically significant reduction in viability of the silenced cells compared to their respective negative controls was found. Interestingly, restoration of cell viability was achieved after incubation of the silenced cells with the exosomes extracted from the untreated cells of the same lines, alone or in the presence of the transfection complex. Wnt signaling was reported to play important functions during embriogenic development and in progression of several types of solid cancers, thus resulting in an excellent therapeutic target [33]. Moreover, Wnt proteins were found at the exosome surface, being responsible for inducing Wnt signaling activity in target cells [34]. The results of the present study seem to support the fundamental role of the Wnt cascade mediated by the FZD10 protein in carcinogenesis, as highlighted by the FZD10-mRNA silencing experiment performed on selected cancer cell lines. A preliminary silencing study was carried out on nine different types of cancer cell lines, originating from the primary tumor (CaCo-2, HLE, Hep-3B, PCL-5 and HLF) or metastatic sites (SW-620, N-87, HUCCT-1, and HGC-27). Interestingly, only the metastatic cells and one among the investigated primary cancer cell lines, namely the HLF cells, showed a significant reduction of proliferation.

Moreover, these cell lines presented a reduction in the expression level of the protein and the FZD10- mRNA, while their subsequent restoration after FZD10-mRNA silenced cell incubation with exosomes were obtained. Very recently, a study carried out on patients affected by colorectal and gastric carcinoma demonstrated that FZD10 is delivered in the bloodstream exclusively by small EVs present in the plasma [8]. Similarly, here, the presence of FZD10 was detected in the exosomes secreted by colon cancer, gastric cancer, hepatocellular carcinoma, and intrahepatic cholangiocellular carcinoma cell lines, which are in total the investigated metastatic cell lines (HGC-27, SW-620, N-87, HUCCT-1 cells), apart from the HLF cells. Therefore, it is reasonable to assume that FZD10, located in the metastatic cells and delivered by the secreted exosomes, plays a crucial role during the cells’ metastatic evolution and their survival. These assumptions are supported by the findings reported by Scavo et al. on the metastatic colon, melanoma and gastric tissues, and the hyper-expression of FZD10 protein in the cytoplasm, as well as on the cell membrane [8]. The results obtained on the primary cancer HLF cell line can be explained considering that this line derives from a poorly differentiated hepatocarcinoma with established mesenchymal features [35]. Also, this hepatocarcinoma cell line is known to present distinctive cancer properties, such as higher proliferative and migratory phenotypes, compared to the well-differentiated hepatocarcinoma cell lines such as Hep-2 and Hep-3B [36]. The regulation of cell proliferation in HLF cells is modulated by different factors like Oct3/4. Indeed, the down-regulation of Oct3 /4 was demonstrated also to regulate proliferation and motility in HLF cells, but not in other hepatocarcinoma cell lines [37], increasing β-catenin protein level with a consequent Wnt signaling amplification, finally enhancing the invasive cellular activity that is characteristic of epithelial-mesenchymal transition [38]. Such features can reasonably explain the behavior in terms of silencing response of HLF compared to the other primary tumor cell lines investigated in this study.

Here, the specific role of exosomes vehiculated FZD10 during carcinogenesis, not only in gastric and colorectal cancer, but also in cholangiocarcinoma and hepatocarcinoma, was demonstrated. Several studies report on the crucial role, in human cancers, of the FZD protein family in regulating cell polarity, proliferation, formation of neural synapses, and other processes [17]; however, to the best of our knowledge, the presence of FZD10 or, in general, of other FZD proteins in EVs, and more specifically in exosomes, was not reported so far. In this study, the ability of exosomes secreted by gastric, colorectal cancer, cholangiocarcinoma and hepatocarcinoma cancer cell lines to deliver FZD10 and restore viability in the silenced cells, that drastically reduced their viability, was *in vitro* demonstrated on HGC-27, SW-620, N-87, HUCCT-1 and HLF cells. Indeed, the role of FZD10 in cancer proliferation for other types of cancer was established, proving, for example, its involvement in the progression of synovial sarcoma by regulating actin reorganization and anchorage-independent cell growth [39]. Moreover, A. Togashi et al. reported that hypoxia-inducible protein-2 (HIG2) binding to FZD10 enhanced oncogenic Wnt signaling and its own transcription, thus suggesting that HIG2-FZD10 interaction and activation of the Wnt signaling pathway are both involved in development and progression of renal cell carcinoma [40].

Our *in vitro* experimental results suggest that FZD10 located in specific cellular compartments, may be transferred in the exosomes secreted by cells and that FZD10 and FZD10-mRNA delivering exosomes may be potential messengers of disease reactivation even in quiescent cells, being FZD10 is directly involved in carcinogenesis and tumor proliferation. Moreover, FZD10 and FZD10-mRNA delivering exosomes can also be thought to be messengers of cellular transformation by transferring of exosomal FZD10-mRNA and to play an active function in long-distance metastatization, as both the pathological protein and, more importantly, its corresponding mRNA, are secreted by the cells involved in the primary tumor constitution through exosomes. 

In addition, FZD10 is indicated to be localized as a cargo at the surface of the exosomes membrane, as suggested by the preliminary results on the immunodetection of FZD 10 by TEM analysis. Such an assumption can be explained, considering the exosomes formation mechanism [41]. According to this mechanism, it can be reasonably guessed that the presence of FZD10 on the membrane of exosomes is due to the cytoplasmic membrane fragments from which the exosomes are generated. Furthermore, since the Wnt presence is well documented in exosomes, it also possible to assume that Wnt proteins and their receptors, such as the FZD10 protein, can be transported through endosomal compartments onto exosomes [34].

These original outcomes pave the way for future studies aiming at elucidation of the role of the Frizzled proteins family and other proteins complex compounds of the Wnt pathway, in exosomes derived from different cell lines, as well as from patients fluids during the carcinogenesis and tumor cells proliferation, and to design innovative drug delivery systems able to target the circulating exosomes on patient fluids for a personalized therapy against cancer development and metastatization.

## Figures and Tables

**Figure 1 cells-08-00777-f001:**
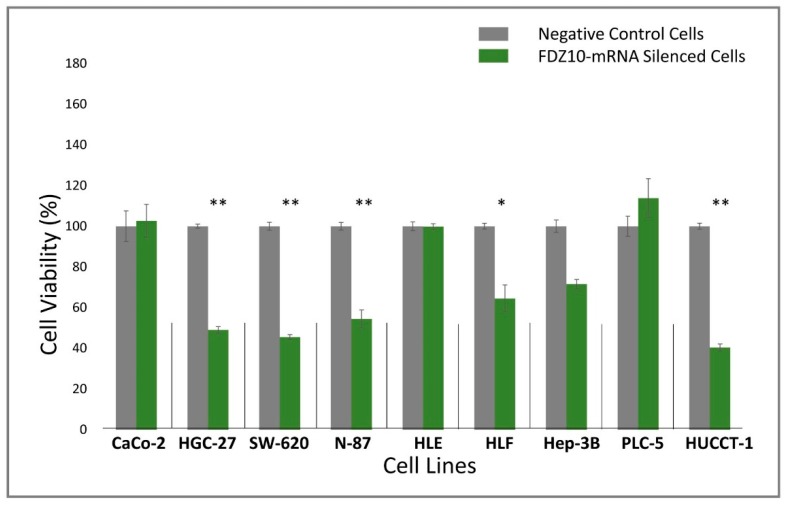
Cell viability, evaluated by MTS cell proliferation assay, of CaCo-2, HGC-27, SW-620, N-87, HLE, HLF, Hep3B, PLC-5, and HUCCT-1 cells after incubation with si-PORT-NeoFX transfection agent and silencer select FDZ10-siRNA, for 96 h. For each cell line, negative control was represented by cells treated only with si-PORT-NeoFX transfection agent. The experiments were repeated three times for each tested cell line. (*) *p* < 0.005, (**) *p* < 0.001 versus negative control.

**Figure 2 cells-08-00777-f002:**
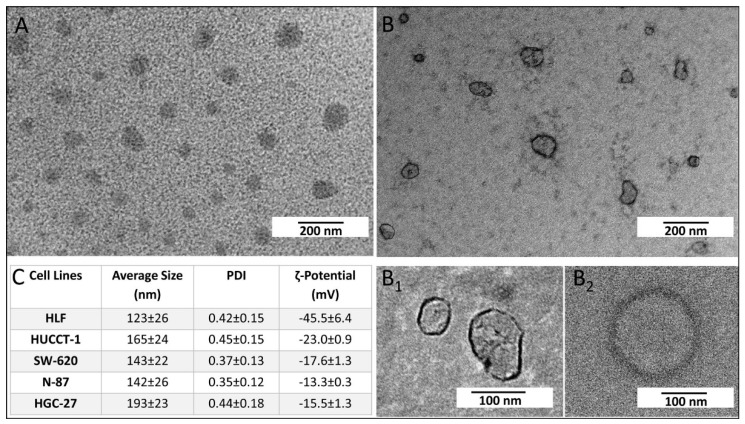
Representative TEM micrographs obtained with positive (**A**) and negative (**B**, **B1** and **B2**) staining of exosomes freshly extracted from culture medium of HUCCT-1 cells. ζ-Potential value, intensity-average hydrodynamic diameter and corresponding polydispersity index (PDI) determined by DLS of the exosomes extracted from culture medium of the untreated HGC-27, SW-620, N-87, HUCCT-1 and HLF cells and suspended in water. Mean ± SD are reported, *n* = 3 (**C**).

**Figure 3 cells-08-00777-f003:**
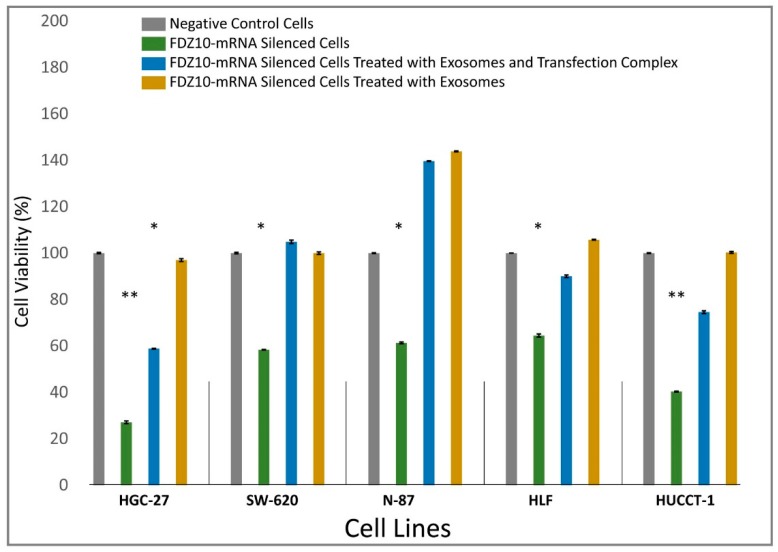
Cell proliferation, evaluated by MTS cell proliferation assay, of HGC-27, SW-620, N-87, HLF, and HUCCT-1 cells after incubation with si-PORT-NeoFX transfection agent and silencer select FDZ10-siRNA and subsequent incubation with a mixture of transfection complex and exosomes or exosomes only. For each cell line, negative control was represented by cells treated only with si-PORT-NeoFX transfection agent. The experiments were repeated three times for each tested cell line. (*) *p* < 0.005, (**) *p* < 0.001 versus negative control.

**Figure 4 cells-08-00777-f004:**
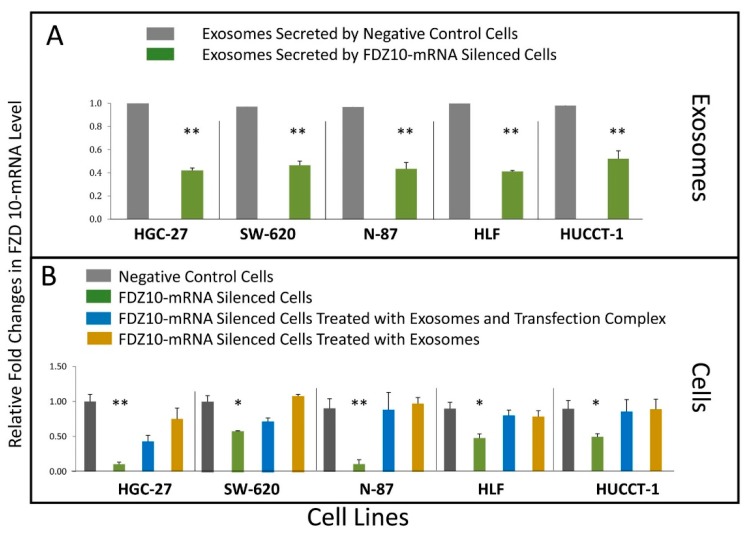
(**A**) Quantitative FZD10-mRNA expression level evaluated by means of real time q-PCR in the exosomes extracted from the culture medium of negative control and FDZ10-mRNA silenced cells, for HGC-27, SW-620, N-87, HLF, and HUCCT-1 cells. Relative quantification analysis by using CFX96 manager from Bio RAD (relative expression software tool). Relative fold changes in FZD10-mRNA expression level (y axis) represent the relative expression of the FZD10-mRNA expression in comparison to the corresponding control (equals to 1 by definition) and normalized by GAPDH housekeeping gene expression. (**B**) Quantitative FZD10-mRNA expression level evaluated by means of real time q-PCR in FZD10 mRNA silenced HGC-27, SW-620, N-87, HLF, and HUCCT-1 cells before and after incubation with a mixture of transfection complex and exosome or exosomes only. For each cell line, negative control was represented by cells treated only with si-PORT NeoFX transfection agent. Relative quantification analysis by using CFX96 manager from Bio RAD (Relative Expression software). Relative fold changes in FZD10 mRNA expression level (y axis) represents the relative expression of the FZD10 mRNA expression in comparison to the corresponding negative control (equals to 1 by definition) and normalized by GAPDH reference gene expression. The experiments were repeated three times for each tested cell line. Data expressed as the mean ± standard deviation (SD), (*) *p* < 0.005, (**) *p* < 0.001 versus negative control.

**Figure 5 cells-08-00777-f005:**
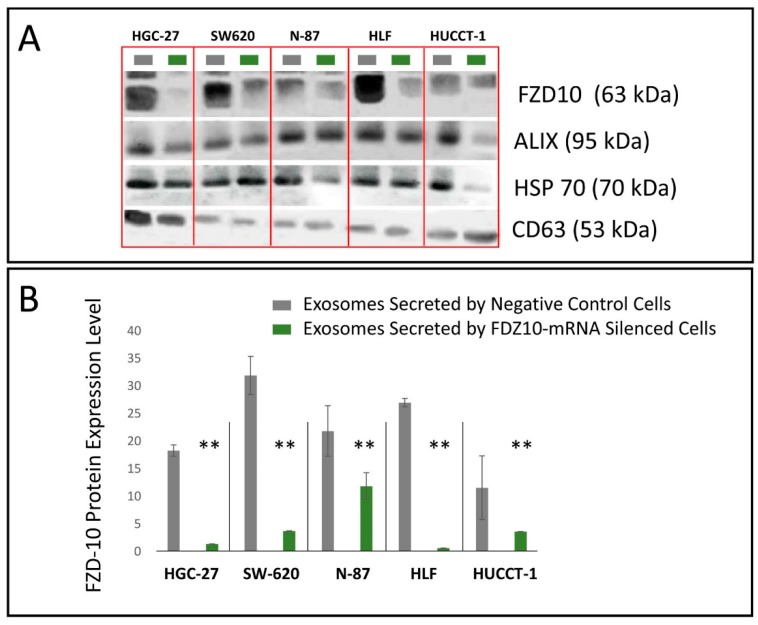
(**A**) Representative Western Blotting of FZD10 protein and three exosomal protein markers (Hsp70, CD-63 and ALIX) and (**B**) semi-quantitative estimation, by densitometry of protein bands, of relative FZD10 expression level in exosomes extracted from the culture medium of negative control and FDZ 10-mRNA silenced cells (HGC-27, SW-620, N-87, HLF, and HUCCT-1 cells), by loading the same total proteins content (20 µg). Molecular mass markers are indicated on the right. For semiquantitative analysis, FZD10 bands are evaluated upon normalization with the corresponding housekeeping HSP-70 protein band, for each sample. (**) *p* < 0.001 versus control, for the exosomes derived from each cell line.

**Figure 6 cells-08-00777-f006:**
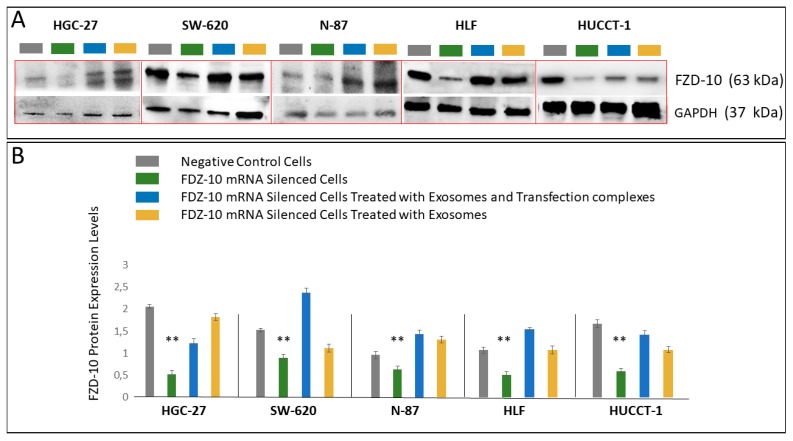
(**A**) Representative Western blotting of FZD10 and GAPDH housekeeping protein and (**B**) semi-quantitative estimation, by densitometry analysis of protein bands, of relative FZD10 expression level FZD10-mRNA silenced HGC-27, SW-620, N-87, HLF, and HUCCT-1 cells, before and after incubation with a mixture of transfection complex and exosome or only exosomes, by loading the same total protein content (20 µg). For each cell line, negative control was represented by cells treated only with si-PORT-NeoFX transfection agent. Molecular mass markers are indicated on the right. For the semiquantitative analysis, FZD10 bands are evaluated upon normalization with the corresponding housekeeping GAPDH protein band, for each sample. (**) *p* < 0.001 versus control.

**Figure 7 cells-08-00777-f007:**
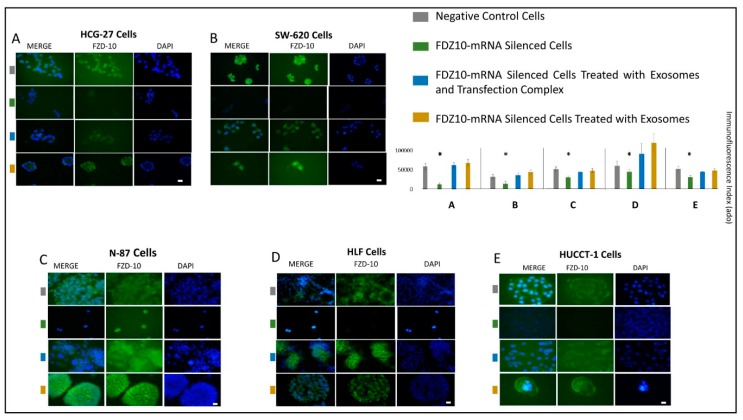
Detection of FZD10 by immunofluorescence confocal microscopy in fixed HGC-27 (**A**), SW620 (**B**), N-87 (**C**), HLF (**D**), and HUCCT-1 (**E**) cells. Confocal microscopy images and immunofluorescence by mean intensity index of FZD10-mRNA silenced cells, FZD10-mRNA silenced cells after incubation with mixture of transfection complexes and exosome and FZD10-mRNA silenced cells after incubation with exosomes only. For each cell line, cells treated only with si-PORT-NeoFX transfection agent were used as negative control. Blue channel: nuclei; green channel: labeled FZD10 and corresponding overlay. Scale bar 20µm. (*) *p* < 0.005.

**Figure 8 cells-08-00777-f008:**
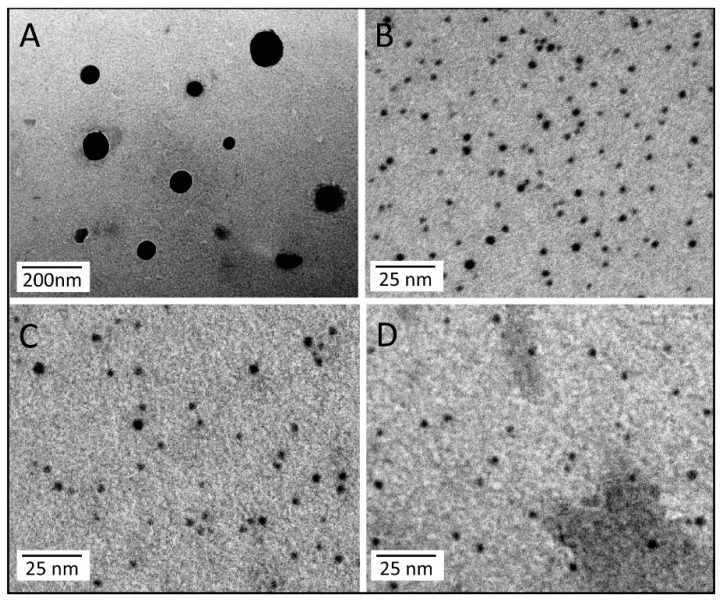
Representative TEM micrographs obtained with positive staining of exosomes freshly extracted from culture medium of HLF cells (**A**), *as synthesized* Au NPs (**B**), *as synthesized* Au NPs after treatment with BSA for 1 h and then with exosomes for 15 min (AuNPs/BSA/Exosomes **C**) and *as synthesized* AuNPs after treatment with exosomes for 15 min (Au NPs/Exosomes, **D**).

**Figure 9 cells-08-00777-f009:**
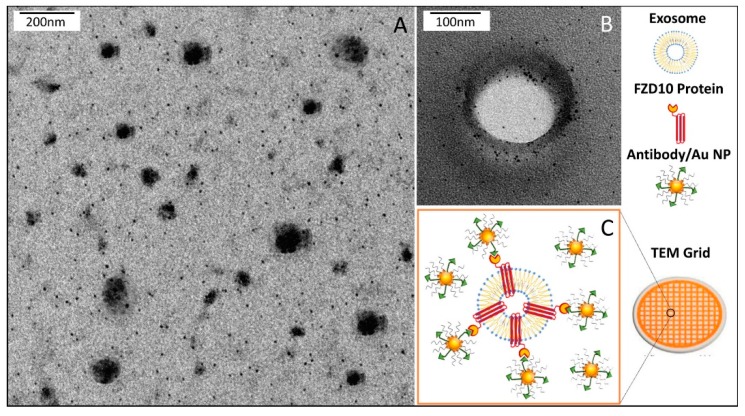
Representative TEM micrographs obtained with positive staining (**A**) and negative staining (**B**) of *as synthesized* Au NPs after treatment with FZD10 antibody for 16 h, with BSA for 1 h, and then with exosomes for 15 min and schematic illustration of the system (**C**).

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
