# Peer review of "FZD10 Carried by Exosomes Sustains Cancer Cell Proliferation"

_cells, 2019, doi:10.3390/cells8080777_

Round 1

Reviewer 1 Report

Summary of the manuscript

The manuscript demonstrated a novel finding that is based on the role of FZD10 (Frizzled 10) & FZD10-mRNA on proliferation in GI cancer cells. In fact, the researchers carried out their experiments on the exosomes of the GI cancer cells where they showed a decrease in cancer cell proliferation with FZD10-mRNA silencing. At the same time, they found a reactivation of the cell viability when they were exposed to the exosomes of the control cell lines.

Major findings

From the cell culture experiments that involved the silencing of FZD10-mRNA, the researchers observed a substantial decline in cell viability on HGC-27, SW-620, N-87, HUCCT-1 and this was promoted by FDZ-10-mRNA silencing.

The MTS data showed that when the exosomes and transfection complex were incubated, all the investigated FDZ-10-mRNA silenced cell lines restored their viability.

They demonstrated a significant reduction in FZD10-mRNA relative fold change in the exosomes extracted from the different cell lines after the silencing experiment. PCR, western blot, and immunofluorescence experiments also validated a marked reduction in FZD10 expression level in the exosomes obtained from the silenced cells compared to those of the exosomes obtained from untreated cells.

So overall, a remarkable reduction in FZD10 and FZD10-mRNA content was found in the silenced cells as well as in their extracted exosomes and hence, it was evident that FZD10 is carried through the exosomes of the cells.

Strength of the manuscript

The manuscript is nicely written, scientifically sound, data presentation are vivid, and so its acceptance is well justified from my side.

Author Response

We are pleased that the work we submitted was overall well considered by the Reviewer, that define our study reported in the manuscript as “nicely written, scientifically sound, data presentation are vivid”, thus recommending its publication on Cells Journal without any revisions.

Reviewer 2 Report

-The "introduction" section should be divided into subsections/paragraphs so as the reader could follow the manuscript. 

-it is not clear why did you select the specific cell lines.

-Appendix A and B should be included in the Materials and Methods Sections. 

 -Please provide information concerning the filters and the lenses that you used with the microscopes.  

-Line 188. It is electron microscopy not "electronic"

-Lines 205-209: which was the p-value that was used in order to consider two groups as statistical significant different?

-Lines 216-221: How do you explain the different cell viability in the different cell lines?

-Line 240: Why you investigate the morphology of exosomes? why is the importance of morphological characteristics of exosomes? Why you did not quantify their morphological characteristics? 

-Line 337-341: How did you quantify the green fluorescence intensity? Was the exposure time the same for all the samples?

-Line 380: These data can be presented as supplementary to support the paper

Author Response

The Reviewer pointed out that only few issues need to be addressed to make the work suited for publication.  We would thank the Reviewer’ for his/her suggestion, and comments have been taken into account in the new version of the manuscript, that we now resubmit for your consideration.

The full list of points arisen by the Reviewer has been answered and discussed point-by-point in the following, and considered in the current version of the manuscript, indicating also the changes eventually made.

Recommendation: Minor revisions

Comments:

-The "introduction" section should be divided into subsections/paragraphs so as the reader could follow the manuscript.

We would like to thank the Reviewer for his/her suggestion, and consequently, in the revised version of the manuscript, the introduction has been divided in three paragraphs: the first paragraph briefly places the study in a broad context, the second one highlights its novelty, while the last one describes the aim of the work pointing out its main conclusions. We hope that the re-phrasing of the “Introduction” section can make it clearer and easier to be followed by the readers.

-It is not clear why did you select the specific cell lines.

We would like to thank the Reviewer for his/her question.

Recently we have started to investigate the role of FZD 10 delivered by EVs in colorectal and gastric carcinoma (8). The present work has intended to extend and deepen the investigation of the role of exosome carried FZD10 the in vitro gastroenteric carcinogenesis and the cells metastatic evolution. For this purpose, 9 cell lines (CaCo-2, HLE, Hep-3B, PCL-5 and HLF) from the primary tumor or metastatic sites (SW-620, N-87, HUCCT-1, and HGC-27) have been taken into account, so as to cover a wide spectrum of gastroenteric cancer types.

Indeed, this aspect has been reported in the “Discussion” section of the manuscript:

“Very recently, a study carried out on patients affected by colorectal and gastric carcinoma demonstrated that FZD10 is delivered in the bloodstream exclusively by small EVs present in the plasma [8]. Similarly, here, the presence of FZD10 was detected in the exosomes secreted by colon cancer, gastric cancer, hepatocellular carcinoma and intrahepatic cholangiocellular carcinoma cell lines, that is in all the investigated metastatic cell lines (HGC-27, SW-620, N-87, HUCCT-1 cells), apart from the HLF cells. Therefore, it is reasonable to assume that FZD10, located in the metastatic cells and delivered by the secreted exosomes, plays a crucial role during the cells metastatic evolution and their survival. These assumptions are supported by the findings reported by Scavo et al. on the metastatic colon, melanoma and gastric tissues and the hyper-expression of FZD10 protein in the cytoplasm, as well as on the cell membrane”

-Appendix A and B should be included in the Materials and Methods Sections.

We would like to thank the Reviewer for his/her suggestion. The content of Appendix A and B, referring to the description of the cell line culture and TEM sample preparation, has been included in the Materials and Methods section of the revised version of the manuscript.

 -Please provide information concerning the filters and the lenses that you used with the microscopes. 

We thank the Reviewer for his/her comment. Specifications of the filters and lenses used for the confocal microscopy analysis have been included in the revised version of the manuscript.

The types of band-pass excitation filters and the magnification used for the investigation by confocal microscopy have been described in the “Materials and Methods” section, by adding the following sentence:

“The images were acquired by using a Kr-Ar and Ar lasers with 488 nm and 358 nm band-pass filters, respectively, for the FZD10 green channel (488 nm) and for the DAPI blue channel (358 nm), at 40x magnification”.

-Line 188. It is electron microscopy not "electronic"

The mistyping error indicated by Reviewer has been corrected in the revised version of the manuscript.

-Lines 205-209: which was the p-value that was used in order to consider two groups as statistical significant different? We would like to thank the Reviewer for his/her comments. Typically, for the evaluation of a statistical significant difference, Kruskal-Wallis, a non-parametric alternative to the one-factor ANOVA test for independent measures test, has been used. On the basis of such an evaluation, we considered statistically significant a difference between the results for each cell line and those of the corresponding untreated cells for either p<0.005 and p<0.001 .

-Lines 216-221: How do you explain the different cell viability in the different cell lines?

The different tested cell lines derived from different human tissues, both metastatic and not metastatic, have been investigated. Namely, HUCCT-1 is an intrahepatic cholangiocarcinoma cell line derived from ascitic fluid, SW620 colon carcinoma cell line from metastatic in lymph node, N-87 stomach metastatic cell line from metastatic site in liver, HGC-27 human gastric metastatic carcinoma line from lymph node and finally HLF hepatocellular carcinoma line from liver. For all the investigated cell lines the same trend in the response to the silencing and the restoring experiments. However, a different extent of the response was found depending on the nature of the investigated cells. Such a different cell viability can be explained considering that the different cell lines derive from different tissues, each characterized by a different proliferative activity and hence a different tumor aggressiveness.

-Line 240: Why you investigate the morphology of exosomes? Why is the importance of morphological characteristics of exosomes? Why you did not quantify their morphological characteristics?

We would like to thank the Reviewer for his/her comments, as they offer us the chance to explain the relevance of the investigation of the morphology of extracted exosomes. Generally, the effectiveness and the validity of the extraction procedure, as well as the quality of the exosomes isolated (from cell culture supernatant or serum) is assessed by using three main techniques, namely DLS, electron microscopy (TEM and/or SEM) and Western blotting. DLS provides average hydrodynamic diameter of the exosomes, thus allowing to determine their average size; TEM (and/or SEM) allows not only to directly measure the diameters of the exosomes, but also to clearly image their morphology. Finally, the Western blot analysis enables the detection of the specific exosomal protein markers. In fact, the results of the morphological characterization of the isolated exosomes have been reported in the manuscript, as TEM, DLS and Western blotting have provided a comprehensive description of their quantitative and qualitative morphological features (average hydrodynamic diameters, size dimensional range, shape, detection of lipid bilayer (Figure 2). Moreover, the presence of the specific exosomal protein markers. ALIX, HSP 70 and CD 63 (Figure 5 A) markers in the total protein contents has been also proven and quantified. Therefore, the combined use of the above mentioned complementary techniques has resulted in a complete investigation of the exosomes, that present features consistent with the previously reported characteristics of exosomes, and therefore allowing to unequivocally recognising as exosomes our extracted vesicles [Zhen Qu et al., Oncotarget, 2017, 8 (46), 80666-80678; Mingtian Wei et al., Oncotarget, 2017, 8 (26), 42262-42271; Rong Xu et al., J. Clin.  Invest.,  2016, 126(4), 1152-1162; J. L. Qu, Digestive and Liver Disease, 2009, 41, 875-880; Maria Principia Scavo et al., Journal of Oncology, 2019, 2715968, 12 pages].

-Line 337-341: How did you quantify the green fluorescence intensity? Was the exposure time the same for all the samples?

We would thank the Reviewer for his/her question.

The fluorescence intensity was determined collecting five images for each sample and analysing them by using the Eclipse Ti2 by Nikon fluorescence microscope and the Interactive software installed on the machine.

The exposure time was set considering the optimal fluorescence imaging conditions for the control cells, and the same time, namely 90 milliseconds, was used for each image acquisition of all the samples. 

In the revised version of the manuscript the following sentence has been now added:

“The fluorescence intensity was quantified by using an exposure time of 90 milliseconds per each acquisition for all the investigated samples.”

-Line 380: These data can be presented as supplementary to support the paper.

We would like to thank the Reviewer for his/her suggestion. Figure S1 and Figure S2 have been included in the revised version of the Supporting Information of the manuscript. In particular, representative TEM micrographs obtained with positive staining of exosomes freshly extracted from culture medium of SW-620, N-87, HCCT-1 and HGC-27 cells and of as synthesized Au nanoparticles after treatment with FZD10 antibody, then with BSA and finally with exosomes extracted from each teste cell line have been introduced.